# Global upper-atmospheric heating on Jupiter by the polar aurorae

J. O'Donoghue[1,2 ✉], L. Moore[3], T. Bhakyapaibul[3], H. Melin[4], T. Stallard[4], J. E. P. Connerney[2,5] & C. Tao[6]

Jupiter's upper atmosphere is considerably hotter than expected from the amount of sunlight that it receives[1–3]. Processes that couple the magnetosphere to the atmosphere give rise to intense auroral emissions and enormous deposition of energy in the magnetic polar regions, so it has been presumed that redistribution of this energy could heat the rest of the planet[4–6]. Instead, most thermospheric global circulation models demonstrate that auroral energy is trapped at high latitudes by the strong winds on this rapidly rotating planet[3,5,7–10]. Consequently, other possible heat sources have continued to be studied, such as heating by gravity waves and acoustic waves emanating from the lower atmosphere[2,11–13]. Each mechanism would imprint a unique signature on the global Jovian temperature gradients, thus revealing the dominant heat source, but a lack of planet-wide, high-resolution data has meant that these gradients have not been determined. Here we report infrared spectroscopy of Jupiter with a spatial resolution of 2 degrees in longitude and latitude, extending from pole to equator. We find that temperatures decrease steadily from the auroral polar regions to the equator. Furthermore, during a period of enhanced activity possibly driven by a solar wind compression, a high-temperature planetary-scale structure was observed that may be propagating from the aurora. These observations indicate that Jupiter's upper atmosphere is predominantly heated by the redistribution of auroral energy.

Jupiter was observed with the 10-m Keck II telescope for five hours on both 14 April 2016 and 25 January 2017 using NIRSPEC (Near-InfraRed Spectrometer[14]), with the spectral slit aligned north–south along the axis of planetary rotation (Fig. 1a). Spectral images were acquired as Jupiter rotated, as shown in Fig. 1b, c, in which rotational–vibrational (ro-vibrational) emission lines of the $H_3^+$ ion can be seen extending from pole to equator. These ions are a major constituent of Jupiter's ionosphere and mainly reside in the altitude range 600–1,000 km above the 1-bar pressure surface[15]. The intensity ratio of two or more $H_3^+$ lines can be used to derive the column-averaged parameters of that ion: temperature, number density and radiance[16]. As $H_3^+$ is assumed to be in quasi-local thermodynamic equilibrium with Jupiter's upper atmosphere[16], its derived temperature is representative of the region. Details of the $H_3^+$ fitting process and global mapping of parameters are provided in the Methods and in Extended Data Figs. 1, 2.

Global maps of upper-atmospheric temperature have been produced previously[17], but the spatial resolution was such that about two pixels covered 45–90° latitude in each hemisphere, making it difficult to assess how the auroral region was connected to the rest of the planet. In those maps, equatorial temperatures were similar to auroral values, a finding that would indicate that a heat source is active at low latitudes. In Figs. 2 and 3, we show near-global maps of Jovian column-averaged $H_3^+$ temperature, density and radiance, which are the product of several thousand individual fits to the spectral data (see Methods). Using a magnetic field model, we have overlaid oval-shaped lines on the polar regions of Jupiter in both Figs. 2 and 3, with each representing the footprint of magnetic field lines that trace from the planet out to a particular distance in Jupiter's equatorial plane[18]. The main (auroral) oval traces on average to $30R_J$ in Jupiter's equatorial plane ($R_J$ is Jupiter's equatorial radius of 71,492 km at the 1-bar pressure level). The satellite footprints of Io and Amalthea are fiducial markers, mapping out from the planet to $5.9R_J$ and $2.54R_J$ in the equatorial plane, respectively.

Temperatures generally decrease from 1,000 K to 600 K between auroral latitudes and the equator, as shown in Figs. 2 and 3. Densities of $H_3^+$, which are enhanced by aurorally driven charged-particle precipitation[19,20], cut off sharply near the main oval on both dates, indicating that the direct influence of the aurora ends within several degrees of the main oval. At the same time, equatorward of the auroral regions, $H_3^+$ temperatures do not sharply fall with latitude. In the absence of any known sub-auroral electric current systems (as are common on Earth[21]) provided through magnetosphere–ionosphere coupling that cause planetary-scale ion–neutral collisions, we interpret the observed temperature gradients as strong evidence that the auroral upper atmosphere is migrating away from the auroral region to lower latitudes and adjacent longitudes, transporting its heat signature along with it. This must then be enabled principally by equatorward-propagating meridional winds.

[1]Department of Solar System Science, JAXA Institute of Space and Astronautical Science, Sagamihara, Japan. [2]NASA Goddard Space Flight Center, Greenbelt, MD, USA. [3]Center for Space Physics, Boston University, Boston, MA, USA. [4]Department of Physics and Astronomy, University of Leicester, Leicester, UK. [5]Space Research Corporation, Annapolis, MD, USA. [6]National Institute of Information and Communications Technology (NICT), Tokyo, Japan. ✉e-mail: jameso@ac.jaxa.jp

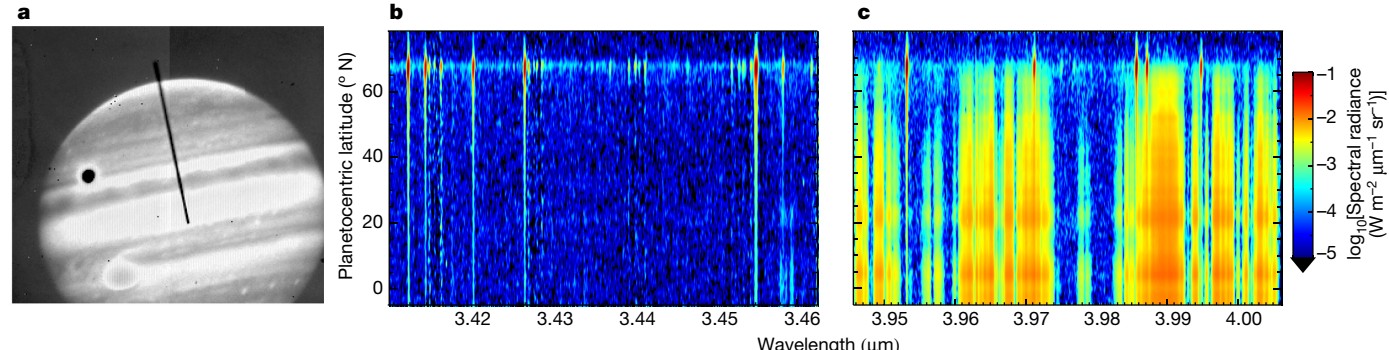

**Fig. 1 | Example set-up showing the acquisition of Jovian spectra on 14 April 2016. a**, Slit-viewing camera image filtered between 2.134 μm and 4.228 μm wavelength. Guide images such as this are taken every 9 s and indicate the slit's position on the sky relative to Jupiter. In this image, the Great Red Spot (bottom left) and satellite Ganymede (top left) can be seen. **b**, **c**, Spectral images of Jupiter showing spectral radiance as a function of wavelength and planetocentric latitude. Most emissions seen in **c** are from the reflection of sunlight from hydrocarbons and hazes. Well-defined vertical lines are $H_3^+$ ro-vibrational emission lines: they are most intense in the polar regions. The R(3,0) and Q(1,0) $H_3^+$ lines at 3.41277 μm and 3.9529 μm are the focus of this study, as their consistently high signal-to-noise (SNR) at all latitudes allows us to map upper-atmospheric energy balance globally. The SNR of $H_3^+$ is high at Jupiter owing to the convenient presence of a deep methane absorption band, particularly in **b** (ref. [27]).

**Fig. 2 | Equirectangular projections of Jupiter's $H_3^+$ column-integrated temperature, density and radiance.** Projections are shown as a function of central meridian longitude (Jovian system III) and planetocentric latitude. Temperature ($T$) and radiance ($E$) panels have uncertainties below 5%, while column densities ($N$) are limited to 20%. Long black-and-white dashed lines show Jupiter's main auroral oval, short black-and-white dashed lines correspond to the magnetic footprint of Io, and the single thick black line corresponds to the magnetic footprint of Amalthea (described in the main text). White indicates regions with no data coverage (or where no results met the uncertainty criteria). Median (and maximum) uncertainty percentiles for 14 April 2016 are: temperature 2.2% (5%), density 9.4% (15%) and radiance 2.2% (5%). Median (and maximum) uncertainties for 25 January 2017 are: temperature 1.6% (5%), density 5.8% (15%) and radiance 1.8% (5%). The Methods describes the mapping process, and Extended Data Tables 1–3 show the bin sizes that were used in each parameter map.

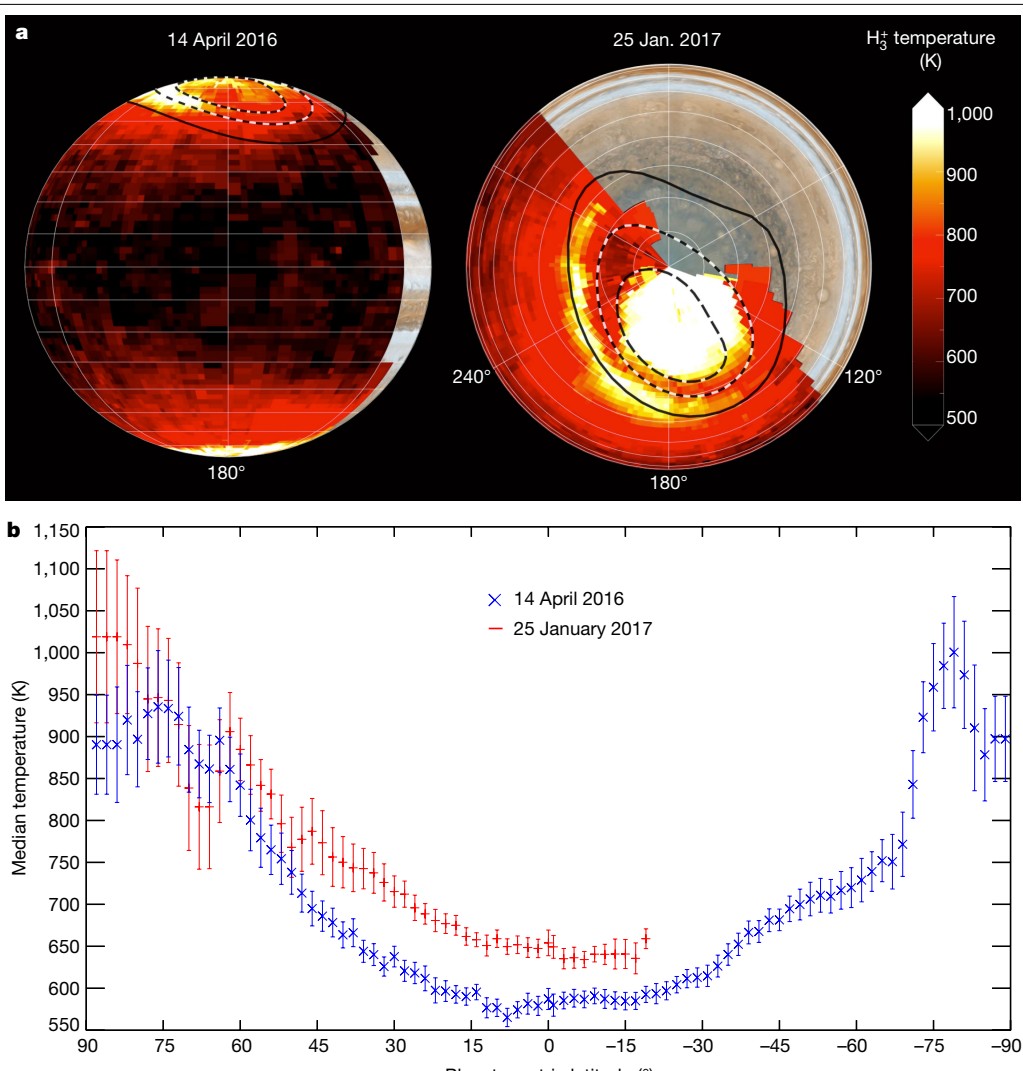

**Fig. 3 | Jupiter's column-averaged H₃⁺ temperatures on 14 April 2016 and 25 January 2017. a**, Orthographic projections of uncertainties in temperature are all below 5%. Long black-and-white dashed lines show Jupiter's main auroral oval, short black-and-white dashed lines correspond to the magnetic footprint of Io, and the single thick black line corresponds to the magnetic footprint of Amalthea (as described in the main text). A visible computer-generated globe of Jupiter based on Hubble Space Telescope imagery is shown underneath the H₃⁺ temperature projection. Image credit: NASA Goddard Space Flight Center and the Space Telescope Science Institute. Note that Jupiter is tilted differently on each date to reveal different features. The longitude and latitude gridlines shown are spaced in 60° and 10° increments, respectively. Median (and maximum) uncertainty percentiles are 2.2% (5%) for 14 April 2016 and 1.6% (5%) for 25 January 2017. **b**, Median Jovian H₃⁺ temperatures found for each latitude across all longitudes. Error bars are 1σ and indicate the variation of temperature over all longitudes. The Methods describes the mapping process, and Extended Data Table 1 shows the spatial bin sizes that were used in each projection.

The Jovian magnetosphere, which is subjected to the solar wind, compresses in response to high dynamic pressure exerted the solar wind[22]. One model has shown that magnetospheric compression events could lead to propagation of heat away from the main auroral oval towards the equator and polar cap, introducing a temporary local temperature increase of 50–175 K (refs. [10,23]). Temperatures were higher planet-wide on 25 January, as were main oval H₃⁺ densities, so a solar wind propagation model[24] was used to examine the solar wind dynamic pressure and other parameters at Jupiter near the dates of our observations. It was found that dynamic pressures were over an order of magnitude higher within a day of the 25 January observations, relative to quiet conditions, and almost three times higher than conditions on 14 April. This is indicated in Extended Data Figs. 3–5 (along with increased activity in other parameters). Total auroral power has previously been found to correlate positively with the duration of quiet solar wind conditions before a solar wind compression[22], so, given the much longer, quieter period of solar wind activity before the 25 January observations reported here

(relative to 14 April), we expect that auroral energy deposition was larger on 25 January. Factoring in the uncertainty of the arrival time of the modelled solar wind at Jupiter, ±1 days on 14 April and ±1.5 days on 25 January, we conclude that Jupiter was observed to be in the middle of a global heating event owing to solar wind compression of the Jovian magnetosphere on 25 January.

An unusual high-temperature structure was found on 25 January equatorward of the main auroral oval, extending for 160° longitude. Here, relatively cold (~800 K) atmosphere is surrounded by hot auroral and sub-auroral atmosphere at ~1,000 K. The structure appears to straddle the fiducial footprint of Amalthea, a region mapping to Jupiter's equatorial plane at 2.5$R_J$ via the magnetic field, but there are no known substantial sources of plasma or current systems linking those regions. It is possible that the structure is a large region of heated upper atmosphere, caught propagating equatorward away from the main auroral oval after a 'pulse' in solar wind pressure was exerted on the magnetosphere[10]. If a heated wave of atmosphere propagates equatorward

from the main auroral oval at similar velocity over all longitudes, it is likely to retain the main oval shape along the way; thus the apparent alignment of the feature with Amalthea may be circumstantial. Here we provide a simple equatorward velocity estimate to examine whether the feature's propagation is realistic. We use the latitude separation between the structure's centre and the main oval, which grows with longitude and therefore with time, since the data are recorded in order of increasing longitude. Equatorward velocities for the hot feature were evaluated between 180° and 260° longitude in steps of 20° longitude, with ~33 min of time elapsing between each step owing to planetary rotation. A median velocity of 620 m s$^{-1}$ was calculated, with minima and maxima of 500 m s$^{-1}$ and 1,500 m s$^{-1}$, respectively. These velocities are similar to equatorward-propagating travelling ionospheric disturbances observed in Earth's ionosphere (300–1,000 m s$^{-1}$)[25] but much higher than equatorward velocities reported at Saturn (up to 100 m s$^{-1}$)[26] and modelled for Jupiter (~180 m s$^{-1}$)[6,10].

In the vicinity of the main oval, $H_3^+$ temperatures and densities are found to anticorrelate. This may be due to charged particles having higher average precipitation energies here relative to other regions, and so penetrating deeper, producing $H_3^+$ at lower, colder altitudes, or evidence that $H_3^+$ is efficiently cooling the atmosphere through infrared emissions[20,27]. Indeed, this may explain how the main oval appears relatively colder relative to adjacent regions, despite the fact the region may have been recently heated by hot structure as it passed by. Alternatively, the hot structure may have been triggered by an event that lasted a short period of time, sending a single wave of hot atmosphere towards the equator, while the main oval returned to relatively quiet conditions. Morphological differences between the aurorae on each date indicate the location and depth of auroral precipitation, which is reflected in the derived parameters, as reported by previous observations[19,20]. The median column-integrated $H_3^+$ densities on 14 April and 25 January between the equator and 30° north were $4 \times 10^{15}$ m$^{-2}$ and $2 \times 10^{15}$ m$^{-2}$, respectively, with the latter being similar to previous values[17]. The $F_{10.7}$ index, an indicator of solar activity via 10.7-cm radio emissions, was 111.8 solar flux units (SFU) and 82.5 SFU on these dates—that is, 36% larger on 14 April, explaining in part this $H_3^+$ density difference. Note that retrieved $H_3^+$ column densities here are expected to be lower by 20% or more of their true value, owing to temperature and density gradients in the upper atmosphere[27]; thus differences in vertical gradients may also contribute to the measured density difference. Radiance maps indicate the degree to which $H_3^+$ radiatively cools the upper atmosphere, and radiance positively correlates with both temperature and density.

Temperature gradients should reveal the dominant heat sources in Jupiter's upper atmosphere, with wave heating showing localized low-latitude peaks[12], and auroral heating showing a monotonic fall from aurora to equator. The gradients presented here are consistent with the latter, at least on these two observed dates. Therefore, the Coriolis forces and other effects that are simulated to confine auroral energy to the magnetic polar regions are evidently overcome at Jupiter. One general circulation model appeared to redistribute auroral heat successfully at Jupiter[6], but subsequent models did not replicate the finding, so the process that allows meridional transport remains unclear[9,10]. At Saturn, latitude–altitude temperature profiles also show a negative gradient from the aurora to lower latitudes[26], while a recent Saturn model presents a possible mechanism to disrupt the trapping of heat in the polar regions there[28]. Main auroral oval $H_3^+$ densities and global $H_3^+$ temperatures were much lower on 14 April than on 25 January, potentially in agreement with model projections[24] that the solar wind dynamic pressure on the Jovian magnetosphere was highest on the latter date, increasing the rates of auroral particle precipitation and global heating[10]. The observations on 25 January also revealed, by chance, a planetary-scale heated structure, which may be propagating away from the main auroral oval in response to a solar wind compression of the magnetosphere, or may originate in the inner magnetosphere via an unknown mechanism.

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

# Article

## Methods

### Additional observing details

On 14 April 2016 and 25 January 2017, Jupiter was recorded between 04:53–10:22 UTC and 11:36–16:28 UTC, respectively. The spectrometer slit measured 24″ long by 0.432″ wide as shown in Fig. 1, and each pixel along the slit had a angular resolution of 0.144″ per pixel. The spectral resolution was $\lambda/\delta\lambda \approx 25,000$. On 14 April, each of the 115 recorded spectral images of Jupiter was 30 s long and formed by six integrations each 5 s long. On 25 January, the 80 recorded spectral images of Jupiter were 60 s long and formed by six integrations each 10 s long. The process of saving spectral images and nodding the telescope between positions results in overhead time which led to an average elapsed time between Jupiter spectra of 2.4 min (14 April) and 3.4 min (25 January), so Jupiter rotated a respective 1.4° and 2.3° in longitude during this time.

### Absolute calibration

For the spectral images, standard astronomical data reduction techniques were applied such as the subtraction of sky spectra from Jupiter spectra to remove unwanted emissions of the Earth's atmosphere (mainly from water), and the accounting of non-uniformity in the response of the NIRSPEC detector via flat fielding and dark-current subtraction. To convert the photon counts at the detector to units of physical flux, a stellar flux calibration was performed using the A0 stars HR2250 and HR3314 for 14 April and 25 January, respectively. This process is outlined in detail in previous studies[16].

### Spatially mapping spectra

Although the width of the slit is 0.432 arcseconds, the longitudes assigned to the slit have a wider range due to atmospheric seeing. The 14 April and 25 January observation nights had clear skies with an atmospheric seeing of 0.61″ and 0.81″, respectively. The use of multiple guider images within each spectral image allowed for tracking errors to be accounted for, such that the derived position of the slit on Jupiter was from the average position of the slit seen in the guiding images. Owing to the width of the slit, atmospheric seeing and the close distances on the planet between each spectral image, multiple spectra can be ascribed to a single longitude × latitude cell (spatial bin). In this work, we use five spatial bin sizes: 10°×10°, 8°×8°, 6°×6°, 4°×4° and 2°×2° longitude × latitude. All data were arranged into five four-dimensional (4D) arrays for each observation night of dimensions longitude × latitude × spectra × overlap. The overlap dimension holds the multiple available spectra of each spatial bin, as displayed in Extended Data Fig. 1. Each of the 4D arrays was collapsed into the three dimensions longitude × latitude × spectra, by taking the median value of each available spectral element. For example, the spectral dimension has 2,048 elements. For a spatial bin that includes 50 overlapping spectra, that means each of the 2,048 spectral elements has 50 values associated with it. By taking the median of the 50 available values, we ensure each spectral element is not skewed towards outlying data. Larger spatial bin sizes encompass more overlapping data, improving the statistical accuracy of the median value obtained, but at the cost of spatial resolution.

### Fitting to $H_3^+$

The $H_3^+$ ion has millions of ro-vibration transition lines that vary in intensity depending on the ion temperature[29], and by finding the ratio between two or more emission lines we can obtain the $H_3^+$ temperature. The total number of emitting ions can then be calculated by dividing the observed emission by that of a single $H_3^+$ ion emitting at the temperature found above, producing a line-of-sight column-integrated density. A cosine function correction of the planetary emission angle is applied to remove the line-of-sight effects of viewing geometry. The radiance of $H_3^+$ (also known as the $H_3^+$ radiative cooling rate) is then found by summing the modelled emission intensities over all wavelengths.

In this work, we used the R(3,0) and Q(1,0) $H_3^+$ lines at 3.41277 μm and 3.9529 μm (respectively) because of their consistently high SNR at all latitudes. These $H_3^+$ lines were fitted to and characterized using MPFIT, a least-squares curve-fitting routine[30], as shown in the example fits of Extended Data Fig. 2. Non-$H_3^+$ emissions were found at some latitudes and were subtracted. The data were then passed to a computational model that determines the parameters of $H_3^+$ based on the line ratios as described by the previous paragraph[16]. Uncertainties in MPFIT and the $H_3^+$ fitting model were propagated through and reflected in the results. Note that these observations are column integrations of the entire path-length of the ionosphere and convolve all vertical structure. Models have demonstrated that these retrieved column-integrated $H_3^+$ densities represent the lower limits of actual values, while column-averaged $H_3^+$ temperatures primarily represent the temperature at the altitude peak of $H_3^+$ density[27].

### Uncertainty-limited mapping of $H_3^+$ parameters

The data in every spatial cell of the five data cubes were fitted so as to produce parameter maps of $H_3^+$ column-integrated temperature, density and radiance, along with corresponding uncertainties. A total of 15 maps were produced for each night: three $H_3^+$ parameters at the five aforementioned spatial bin sizes. Ideally this study would use only the 2°×2° maps, but these smaller bins can have lower SNR outside of the hot auroral regions and thus undesirably high uncertainties. In such a case, selecting a larger 4°×4° bin size to gather more signal and reduce uncertainties is preferable, even though it reduces our ability to see fine detail spatially. To produce a map populated by low-uncertainty data at the smallest bin sizes possible planet-wide, we introduce a technique called uncertainty-limited binning. For example, an $H_3^+$ temperature map is produced by starting with a blank map, and then all 2°×2° longitude × latitude resolution temperatures that have uncertainties under 5% are added. For parts of the map that were not populated by this first pass, data are drawn from the next spatial size up—the 4°×4° temperature map (again with uncertainties under than 5%)—and this process is then iterated for all remaining larger spatial bin sizes up to 10°×10°. $H_3^+$ column-integrated temperature and radiance maps are uncertainty-limited to 5%, while densities are limited to 15%.

### Data availability

Observational data that are the basis of this study are publicly available on the Keck telescope observatory archive at https://koa.ipac.caltech.edu/cgi-bin/KOA/nph-KOAlogin?more under Semester search terms '2016A' or '2017A' and Principal Investigator 'ODonoghue'. Source data are provided with this paper.

### Code availability

Open-source computer code used for fitting to $H_3^+$ and producing temperatures, densities and radiances is available in the Python programming language at https://pypi.org/project/h3ppy/. Least-squares curve-fitting routine MPFIT is referenced in the Methods and available as part of the Interactive Data Language (IDL) suite of available programmes at https://www.l3harrisgeospatial.com/docs/mpfit.html.

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

**Acknowledgements** The data presented herein were obtained at the W. M. Keck Observatory, which is operated as a scientific partnership among the California Institute of Technology, the University of California and NASA. The Observatory was made possible by the generous financial support of the W. M. Keck Foundation. The authors wish to recognize and acknowledge the cultural role and reverence that the summit of Maunakea has always had within the indigenous Hawaiian community. We are fortunate to have the opportunity to

conduct observations from this mountain. L.M. was supported by NASA under grant no. NNX17AF14G issued through the Solar System Observations Program and grant no. 80NSSC19K0546 issued through the Solar System Workings Program.

**Author contributions** J.O'D. collected, analysed and interpreted the data, and wrote the paper. L.M. greatly assisted in collection and reduction of data and interpretation of the results. T.B. assisted in key data reduction. H.M. provided computer code necessary for the analysis of data. H.M. and T.S. assisted in data collection, analysis and interpretation. J.E.P.C. provided code to map Jupiter's magnetic field and assisted in the interpretation of data. C.T. provided solar wind simulation results. All authors provided comments on the manuscript.

**Competing interests** The authors declare no competing interests.

**Additional information**
**Correspondence and requests for materials** should be addressed to J.O'D.

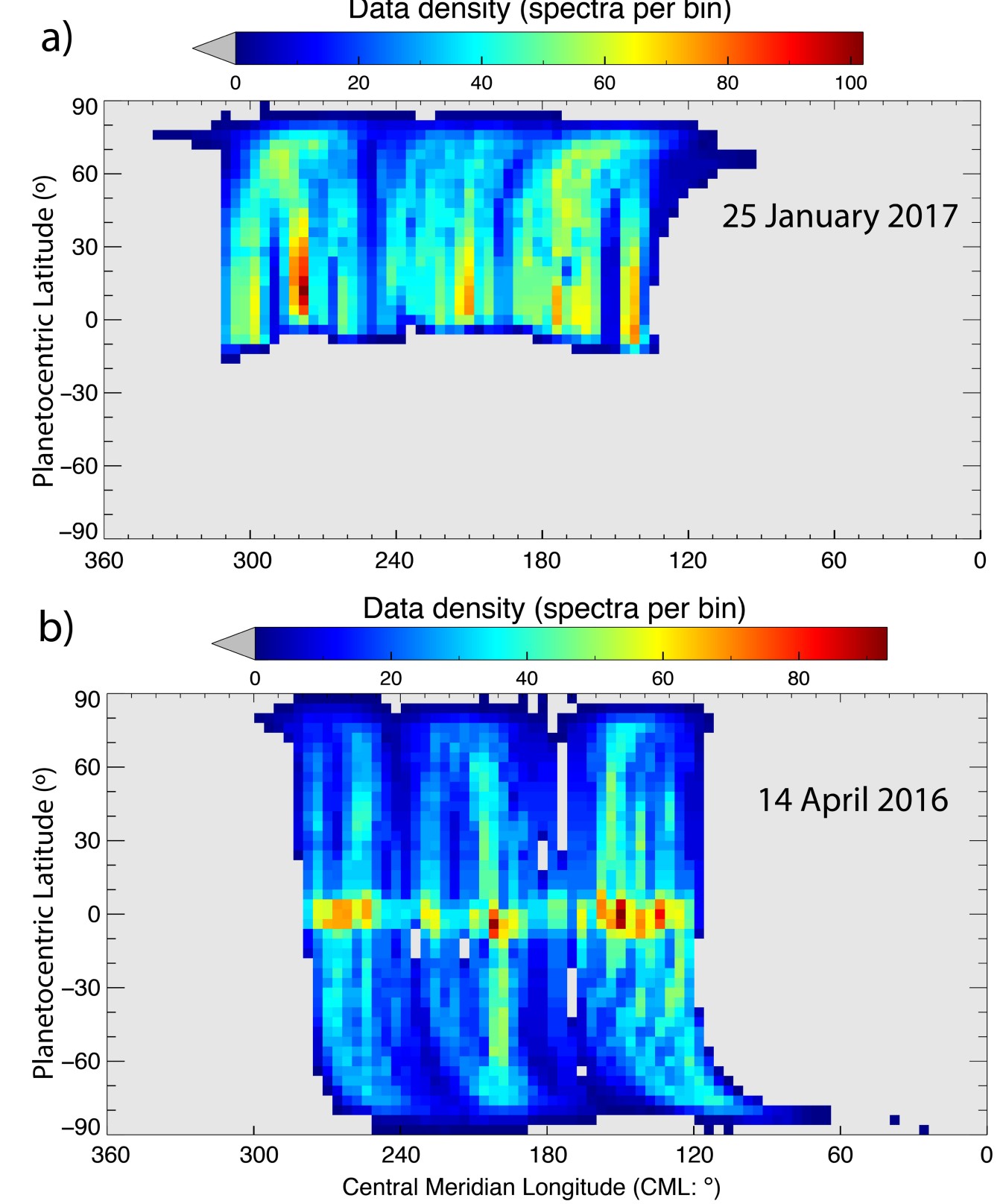

**Extended Data Fig. 1 | Data coverage (density) at 4° × 4° longitude × latitude resolution. a**, 25 January 2017; **b**, 14 April 2016. Each element contains an array of intensities as a function of wavelength, as described in the main text. Data cubes like this also exist for spatial resolutions 2° × 2°, 6° × 6°, 8° × 8° and 10° × 10°.

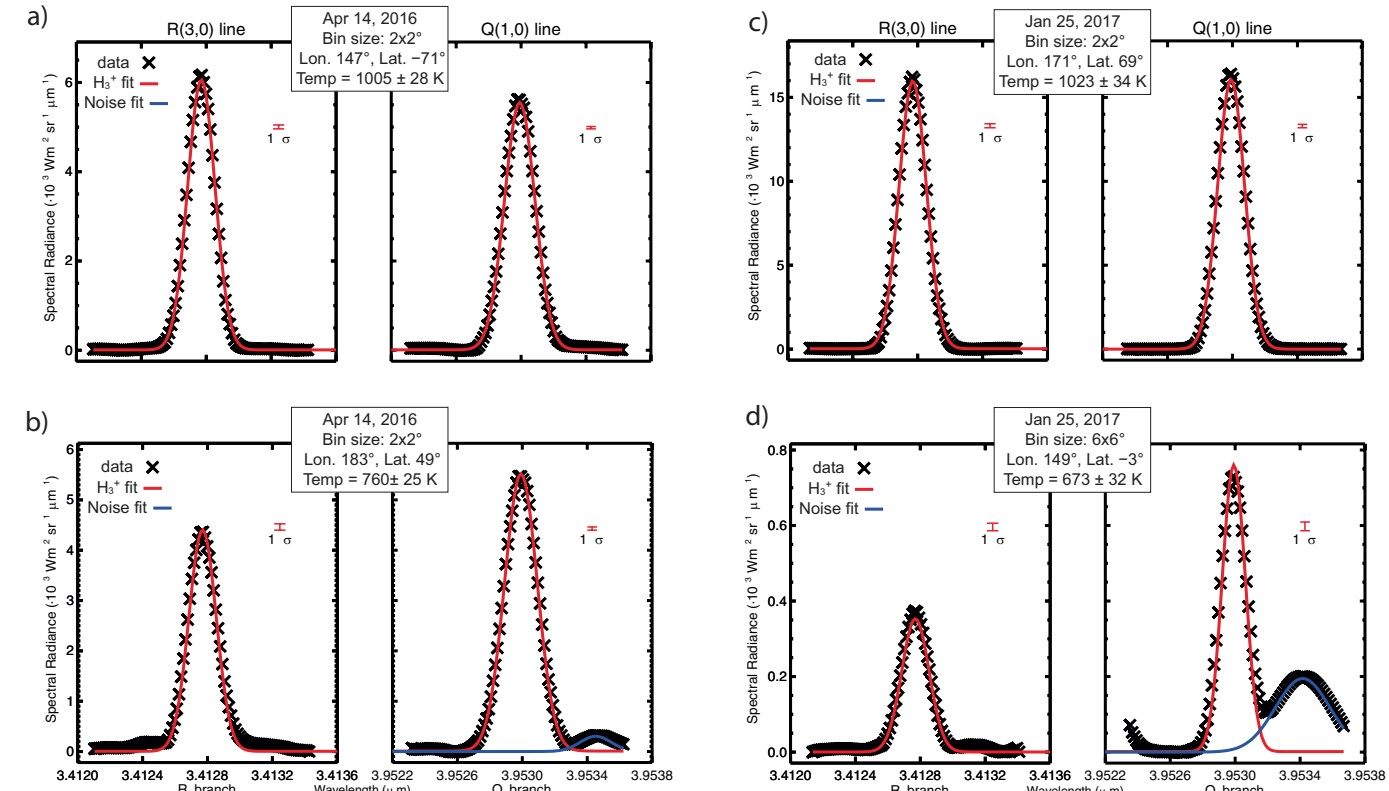

**Extended Data Fig. 2 | Selected example fits to data representing multiple days, differing longitudes, latitudes and spatial bin size. a,b,** 14 April 2016; **c,d,** 25 January 2017. Red lines, fits; black crosses, data. Fits to non-$H_3^+$ emissions are denoted as noise and $1\sigma$ uncertainties to each line are indicated.

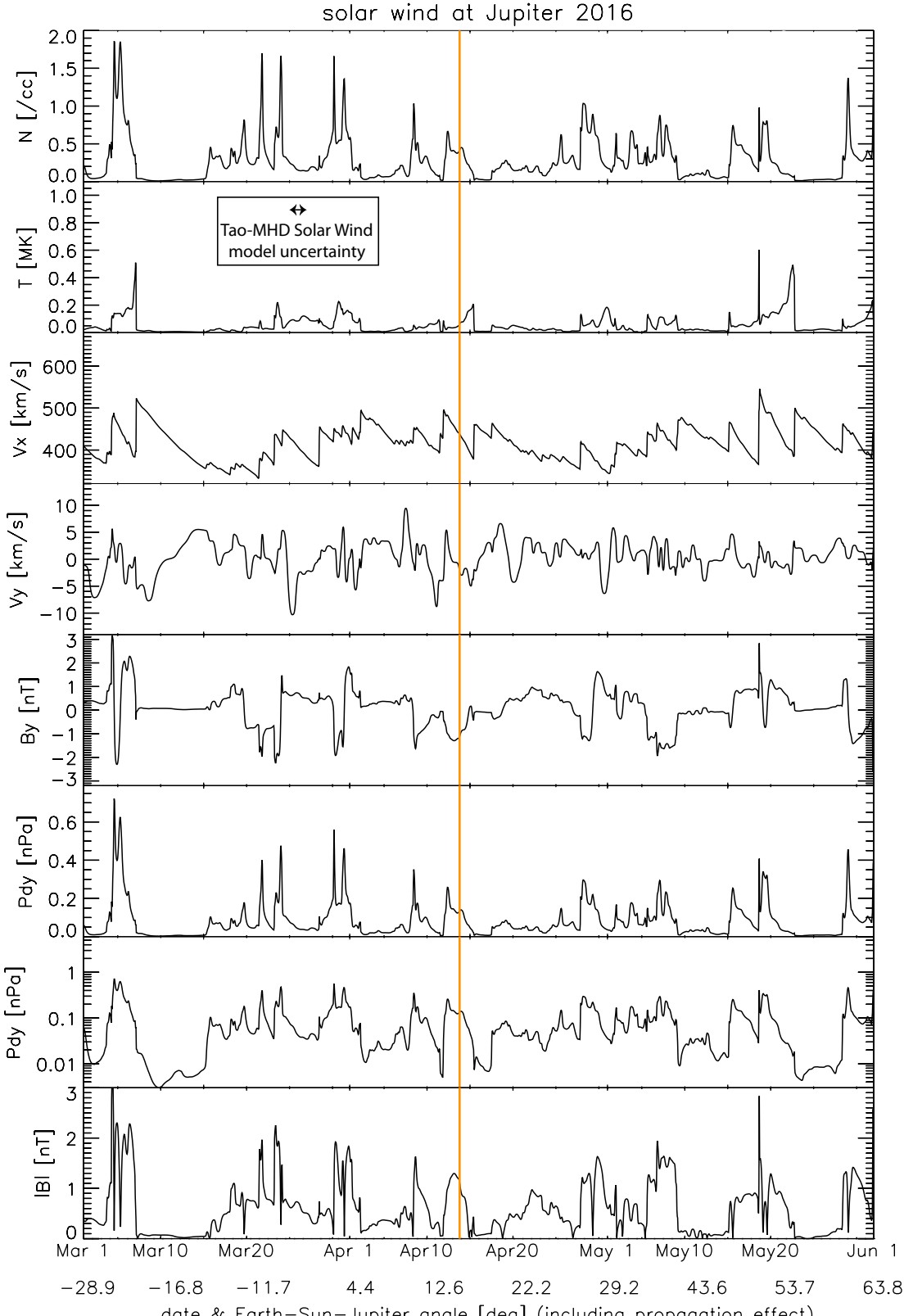

**Extended Data Fig. 3 | A 1D model of solar wind propagation surrounding the dates of the observations reported here.** The orange shaded regions mark the period of ground-based observations. From top to bottom, each panel corresponds to solar wind density, temperature, radial (*x*; Jupiter–Sun line) and azimuthal (*y*; direction of planetary orbital motion) velocities, magnetic field $B_y$, dynamic pressure of the solar wind plotted linearly and logarithmically, and the absolute magnetic field magnitude $|B|$ (ref. [24]). The 1σ uncertainty in arrival time of the solar wind at Jupiter is denoted by the horizontal, arrowed lines. Produced using Tao–MHD, the magnetohydrodynamic model by Tao et al.[24].

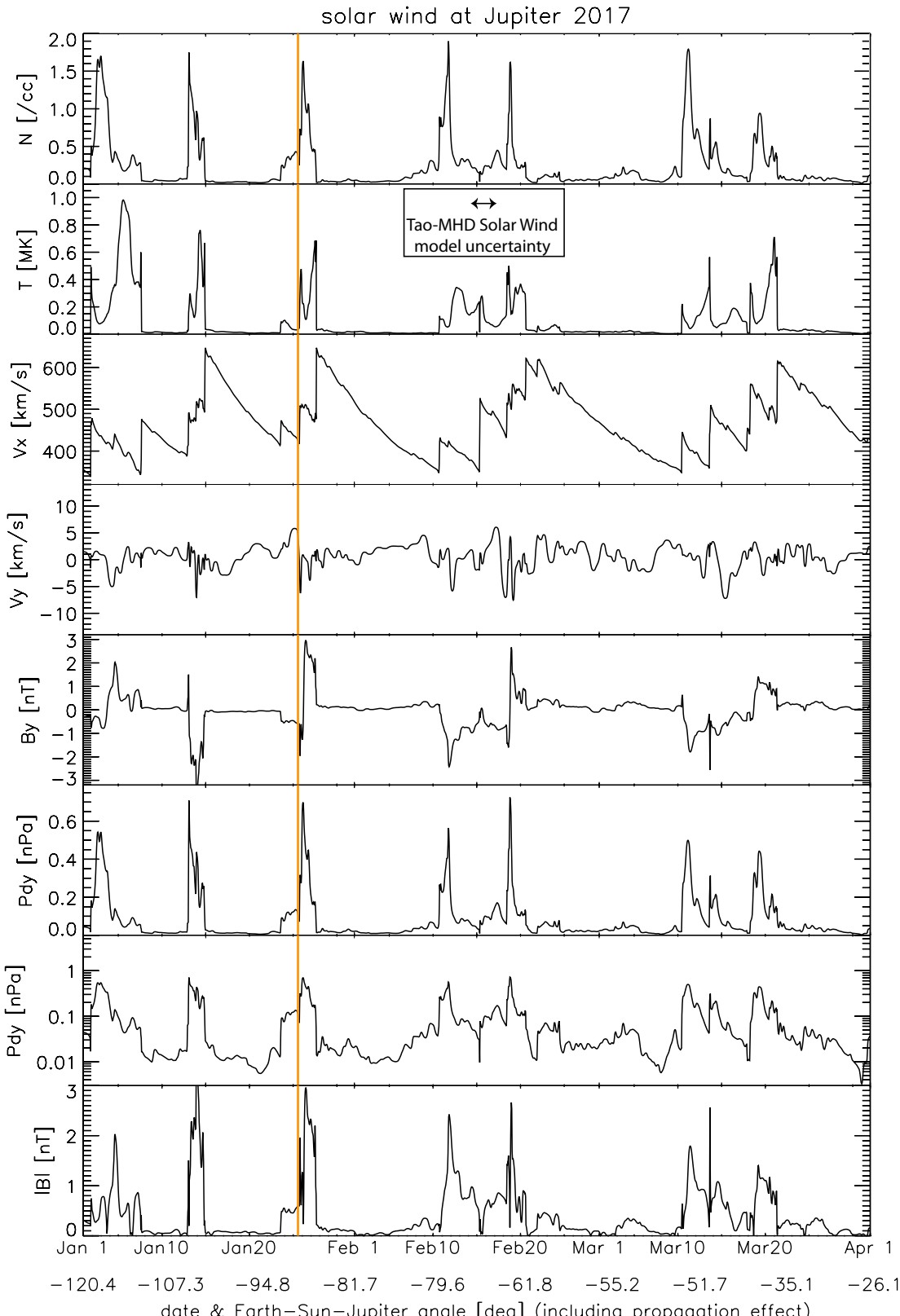

**Extended Data Fig. 4 | A 1D model of solar wind propagation surrounding the dates of the observations reported here.** The orange shaded regions mark the period of ground-based observations. From top to bottom, each panel corresponds to solar wind density, temperature, radial (*x*; Jupiter–Sun line) and azimuthal (*y*; direction of planetary orbital motion) velocities, magnetic field $B_y$, dynamic pressure of the solar wind plotted linearly and logarithmically and the absolute magnetic field magnitude $|B|$ (ref. [24]). The $1\sigma$ uncertainty in arrival time of the solar wind at Jupiter is denoted by the horizontal, arrowed lines.

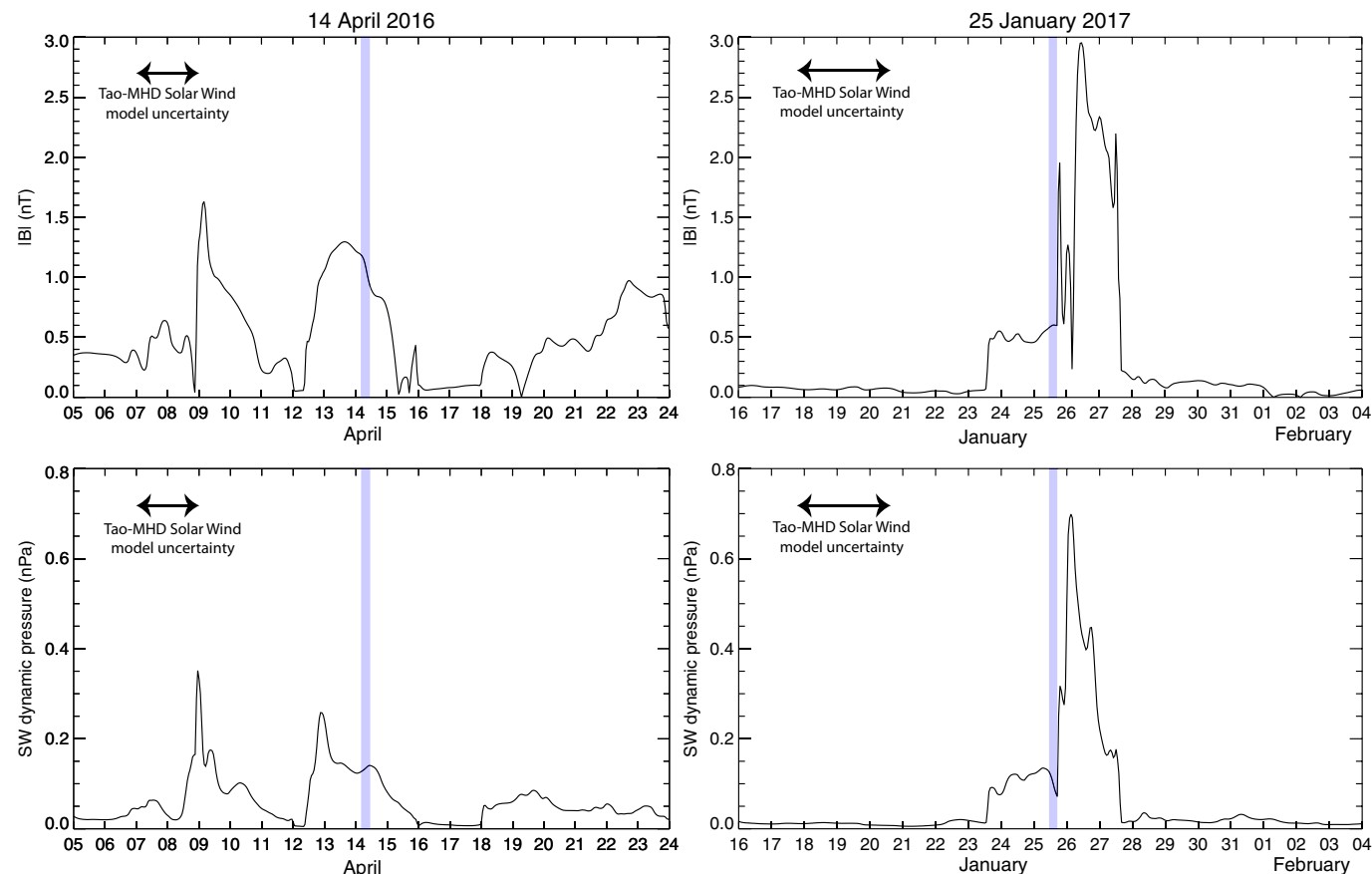

**Extended Data Fig. 5 | A 1D model of solar wind propagation closely surrounding the dates of the observations reported here.** The blue shaded regions mark the periods of ground-based observations. Absolute magnetic field magnitude |*B*| and dynamic pressure of the solar wind (SW) at Jupiter are shown as a function of date and time[24]. The 1σ uncertainty in arrival time of the solar wind at Jupiter is denoted by the horizontal arrowed lines.

**Extended Data Table 1 | Example distribution of 5% uncertainty-limited temperature results in Fig. 2 and Fig. 3a, d by bin size**

| Bin Size<br>Lon. × Lat. | 14 April 2016<br>Cells (% total) | 25 Jan. 2017<br>Cells (% total) |
|---|---|---|
| 2°× 2° | 5166 (65.2%) | 3761 (72.4%) |
| 4°× 4° | 1477 (18.6%) | 499 (9.6%) |
| 6°× 6° | 742 (9.4%) | 333 (6.4%) |
| 8°× 8° | 308 (3.9%) | 255 (4.9%) |
| 10°× 10° | 228 (2.9%) | 348 (6.7%) |
| Total: | 7921 (100%) | 5196 (100%) |

Each cell of Figs. 2 and 3 is of dimension 2° × 2° longitude × latitude, with the majority of data in each of them being sourced from the 2° × 2° bin size. Uncertainties are 1σ.

**Extended Data Table 2 | Example distribution of 20% uncertainty-limited column-integrated density results in Fig. 2b, e by bin size**

| Bin Size Lon. × Lat. | 14 April 2016 Cells (% total) | 25 Jan. 2017 Cells (% total) |
|---|---|---|
| 2°×2° | 4193 (54.1%) | 3652 (70.5%) |
| 4°×4° | 1509 (19.5%) | 555 (10.7%) |
| 6°×6° | 1201 (15.5%) | 338 (6.5%) |
| 8°×8° | 491 (6.3%) | 284 (5.5%) |
| 10°×10° | 363 (4.7%) | 352 (6.8%) |
| Total: | 7757 (100%) | 5181 (100%) |

Each cell of Fig. 2 is of dimension 2° × 2° longitude × latitude, with the majority of data in each of them being sourced from the 2° × 2° bin size. Uncertainties are 1$\sigma$.

**Extended Data Table 3 | Example distribution of 5% uncertainty-limited radiance results in Fig. 2c, f by bin size**

| Bin Size<br>Lon. × Lat. | 14 April 2016<br>Cells (% total) | 25 Jan. 2017<br>Cells (% total) |
|---|---|---|
| 2°× 2° | 5890 (73%) | 3890 (74.1%) |
| 4°× 4° | 1049 (13%) | 455 (8.7%) |
| 6°× 6° | 542 (6.7%) | 321 (6.1%) |
| 8°× 8° | 283 (3.5%) | 258 (4.9%) |
| 10°× 10° | 303 (3.8%) | 327 (6.2%) |
| Total: | 8067 (100%) | 5181 (100%) |

Each cell of Fig. 2 is of dimension 2° × 2° longitude × latitude, with the majority of data in each of them being sourced from the 2° × 2° bin size. Uncertainties are 1σ.