## [Peer Review File · Nature]

Manuscript Title: Global upper-atmospheric heating at Jupiter by the polar auro- rae

Redactions – Mention of other journals

This document only contains reviewer comments, rebuttal and decision letters for versions considered at *Nature*. Mentions of the other journal have been redacted.

Reviewer Comments & Author Rebuttals

Reviewer Reports on the Initial Version:

Referees' comments:

Referee #1 (Remarks to the Author):

The paper 'Global heating of Jupiter's upper atmosphere by auroral energy circulation' by O'Donoghue and colleagues presents new measurements of the Jovian atmosphere obtained with NIRSPEC at the Keck II telescope in April 2016 and January 2017. Auroral emissions are investigated to derive H₃⁺ ion concentration and temperature through intensity ratio of H₃⁺ lines. An increased temperature is observed in case of the reported measurements in January 2017 that is explained with an excess of auroral energy distributed in the upper atmosphere. The study proposed in the manuscript is new and interesting for the astronomical community. It allows to advance in the so-called 'giant planet energy crisis', presenting new insight on the ionosphere-magnetosphere coupling at Jupiter.

The paper is well written and clearly explain the observations and analysis.
The paper is suitable for publication on the Journal.

However, there are few points that are missing in my opinion.

A comparison with other measurements, for example with the Juno observations in a period close to the ones reported in the paper, would be helpful in getting a wider picture of the Jovian atmosphere. For example Juno measurements during the 4th passage on 1st February 2017 would be useful to check if a similar temperature enhancement is still present few days after the data obtained by the authors.

I would also suggest to discuss the JTGCM model results (Bougher et al. 2005, JGR 110, doi:10.1029/2003JE002230) and compare the observed features to this model. It shows that the observed Jovian temperatures and densities in the thermosphere can be satisfactorily matched with a combination of auroral particle and Joule heating at low and high latitudes.

Sentence at line 111 is confusing: temperatures on 25 January are higher inside the main oval with a cold region (T about 650 K) outside the main oval at longitudes 240-330°. Please better describe this part or rephrase.

In Figure 3 a and d, the authors show the temperature maps obtained in the two dates. It is quite intriguing the fact that temperature is maximum inside the main oval and in the longitude region 90-240° on 25th January, while on 14th April the maximum temperature value is outside the main oval and at 270-300° longitude. A different behavior seems to occur on the Southern hemisphere

during the same date, where there is a hot region at about 150°. It would be interesting to discuss in the text these differences among the two dates.

Figure 6 (Extended data Figure 3) shows the H3+ temperature as a function of latitude obtained during the two nights of observations. Although it is clear that on 25th January the retrieved temperature is higher than on 14th April for latitudes of 45° equatorward, error bars are missing and so it is not completely evident if the difference is significant. Another possibility would be showing the temperature difference, associated with the error bars superposed to the temperature. This would help in immediately catching the amount of temperature difference.

Referee #2 (Remarks to the Author):

General comment

This study addresses the important question of the heat source that maintains the high temperature observed in the Jovian upper atmosphere. This 'energy crisis' has been feeding research and controversies in the scientific community for ten of years. It was clearly demonstrated that ultraviolet solar radiation, the main heat source of telluric upper atmospheres is insufficient by hundreds of Kelvin to warm up Jupiter's thermosphere. One of the most obvious proposed alternatives was that auroral heating through Joule heating and collisional heating in the high-latitude regions was the missing source. This idea was confirmed following the first quantitative observations by HST and the subsequent estimates of the global auroral power deposited in both hemispheres.

This rather straightforward deduction was challenged when 3-D Jovian GCMs indicated that the meridional heat transfer to lower latitudes was ineffective owing to the strong Coriolis force and did not provide the heat necessary to explain the observed mid- and low-latitude temperatures.

In this study, the authors report ground-based spectral measurements of the thermospheric temperature measured on two occasions. They claim that the strong latitudinal temperature gradient they measured and the presence of a region of enhanced temperature separated from the auroral oval region both demonstrate that the heat transfer from high to low latitudes is efficient.

This is a report of high quality ground-based observations made with the 10-meter Keck II telescope. It clearly shows the presence of an exospheric temperature gradient between the auroral regions and low latitudes in both hemispheres. The importance of auroral heat transfer is further supported by the differences observed between two observation days, one of them possibly concurrent with an increase in the solar wind activity. The data analysis is careful, including a full analysis of the statistics of the measurements. The text is well written and the illustrations are quite clear and well documented.

The statistics is reliable and the error bars are clearly defined.

My main concern is about the novelty and the impact of these results:

- Unfortunately, no parallel observation of the auroral morphology from HST or Juno could be used to confirm or not an intensification of the auroral precipitation in January 2017
- Previous observations, with coarser spatial resolution, have already indicated that the mid- and low-latitude temperatures are lower than those in the auroral region. See for example, Johnson et al. (JGR, 2018), Moore et al. (GRL, 2017). However, the characteristics of the latitudinal gradient are best seen in Extended data figure 3 of this study
- no mention is made of infrared observations made with the JIRAM on board Juno such as the analysis by Dinelli et al. (2019, Phil. Trans. R. Soc. A 377: 20180406). They offer a spatial resolution between 45 and 130 km.

- Many analogies are made with observations or models of Saturn, which has a different behavior in many respects, although they could refer to Jupiter studies instead.

Given these restrictions, I leave it to the editor to decide whether this study should be published in Nature or another journal such as **[redacted]**.

Specific comments

First paragraph:

l. 9-11 : the lower temperature at low latitude compared to the auroral values is not a new characteristic. It has been observed and discussed before.

l. 11-14 : The sentence 'During a periodregion' is very long. I suggest dividing it into two separate sentences.

l. 30-32 : Juno has revealed that the source of precipitation may be complex and is generally NOT associated with signatures of field aligned currents (Mauk et al., 2019; Bonfond et al., JGR, 125, e2020JA028152).

There are many references better suited establishing that electron precipitation heats up the Jovian upper atmosphere.

l. 70 : there are many better references in the literature to H₃⁺ density increase in Jupiter's aurora rather than Saturn's case (ground-based, Juno JIRAM).

l. 105-108 : How do you determine the 'start time' of the heat wave equatorward propagation?

l. 108-110 : what conclusion do you derive from the discrepancy between the Jovian, Saturnian and terrestrial velocities ?

l. 114 : do you mean in the auroral oval ? How do these values compare with column density measurements reported in the literature?

l. 132: this is a very definite statement considering the uncertainties.

References: some references refer to Saturn's aurora although direct reference to Jupiter's case would be more appropriate.

Extended data figure 3: this figure appears as a key result of this study and deserves to be included in the main manuscript. If space limitations apply, I would suggest making it an additional panel of Figure 2.

Referee #3 (Remarks to the Author):

This paper presents new observations of H₃⁺ densities and temperatures across the disk of Jupiter. The results clearly indicate temperatures decreasing from the auroral regions toward the equator, strongly suggesting that auroral heating is an important source for Jupiter's globally high thermospheric temperatures. During the second of two observations, a remarkable detached arc of higher temperatures was observed in the northern hemisphere, centered on an M-shell of ~2.5 (approximately the orbit of Amalthea). These data are stunning, and are worthy of publication in Nature on their own. However, the authors include some speculative

interpretations of the data in at least three cases (listed below) which are not justified in the current text. These should either be defended with suitable calculations, or the claims should be watered down a lot.

- 1) One instance of this is in trying to link the arc (and generally higher) temperatures seen in the second observation to solar wind variations, without any real justification as to how the solar wind might influence the upper atmosphere in a region connected to the inner magnetosphere. It has been demonstrated in previous studies that most of Jupiter's auroral activity is internally generated, although solar wind variations certainly play a part in modifying polar auroras. To support their idea that solar wind activity is connected to the high temperatures observed at sub-auroral latitudes, it would be important to demonstrate some physical mechanism linking these two vastly separated regions.
- 2) Another example is the derivation of a meridional speed for the arc feature by assuming that the feature is moving coherently equatorward and began at the main auroral oval. I can understand the interest in deriving something about the evolution of this interesting feature from the snapshot observation, but it seems entirely plausible that it isn't moving at all. The arc feature lies pretty much along a single M-shell, which would be expected if it had a magnetospheric origin. If it were moving equatorward from the main oval, it seems hard to imagine why the atmosphere it has already passed over at higher latitudes would cool off to the same levels that are seen at lower latitudes that haven't been reached yet.
- 3) Finally, the assumption that the Coriolis barrier to equatorward transport can be overcome by Rayleigh friction, as speculated for Saturn in the Müller-Wodarg et al. (2019) paper, is a very interesting idea, but for this paper it should at least be tested for plausibility. The Rayleigh friction acts like viscous drag to damp zonal winds, but winds in Jupiter's auroral upper atmosphere are observed by Doppler-shifted H_3^+ emissions to be very large – on the order of km/s. Please provide some estimates for the Rayleigh friction profiles that could allow meridional transport while not shutting down the auroral electrojet.

If the authors could address these major concerns, then I would support publication in Nature.

Minor comments (mostly to improve readability):

1.2 – “Magnetosphere-atmosphere” -> “On Jupiter, magnetosphere-atmosphere”

1.4 – “global circulation” -> “thermospheric global circulation”

1.6 – “insufficient” -> “weak”

1.7 – “warm equatorial” -> “warm equatorial thermosphere”

1.18 – “was recorded by” -> “was observed with”

1.30 – “electrons precipitate” -> “energetic electrons precipitate”

1.31 – “into the planet” -> “toward the planet”

1.32 – “intense auroral emissions” -> “intense auroral emissions through impact”

excitation of ambient H₂ molecules”

1.33 – “constrast” -> “contrast”

1.34 – “based on solar heating” -> “based on only solar heating”

1.37 – “shown that heat within auroral is” -> “found that heat within the auroral zone is”

1.40 – “altitude⁷” In general, I object that many of the references are to derivative studies, rather than the original work or major reviews. This is just one particular case, but wave heating studies have a long history before this reference.

1.47 – “by the Cassini spacecraft” -> “by Cassini UVIS stellar occultation results”; while discussing other giant planets, you might mention that Neptune also has a large thermosphere temperature, but it has very weak auroras.

1.66 – “pressure surface” -> “pressure level”

1.73 – “sub-auroral electric current systems” -> “sub-auroral electric current systems (as are common on Earth)”

1.107 – “median velocity of 620 ms⁻¹”; as discussed above, if I understand what you’ve done here, it seems that you’ve just measured the latitude difference as a function of longitude between the M-shells of 2.5 and 30.

1.114 – “The F_{10.7} index”; does the H₃⁺ auroral brightness vary with F_{10.7}?

Author Rebuttals to Initial Comments:

The authors and I thank you kindly for your time in reviewing our manuscript, especially during this global pandemic. The paper is now much improved because of your inputs.

Referees' comments:

Referee #1 (Remarks to the Author):

The paper ‘Global heating of Jupiter’s upper atmosphere by auroral energy circulation’ by O’Donoghue and colleagues presents new measurements of the Jovian atmosphere obtained with NIRSPEC at the Keck II telescope in April 2016 and January 2017. Auroral emissions are investigated to derive H₃⁺ ion concentration and temperature through intensity ratio of H₃⁺ lines.

An increased temperature is observed in case of the reported measurements in January 2017 that is explained with an excess of auroral energy distributed in the upper atmosphere.

The study proposed in the manuscript is new and interesting for the astronomical community. It allows to advance in the so-called ‘giant planet energy crisis’, presenting new insight on the ionosphere-magnetosphere coupling at Jupiter.

The paper is well written and clearly explain the observations and analysis. The paper issuitable for publication on the Journal.
Thank you for reviewing our manuscript.

However, there are few points that are missing in my opinion. A comparison with other measurements, for example with the Juno observations in a period close to the ones reported in the paper, would be helpful in getting a wider picture of the Jovian atmosphere. For example Juno measurements during the 4th passage on 1st February 2017 would be useful to check if a similar temperature enhancement is still present few days after the data obtained by the authors.

We have now referenced Juno JIRAM H3+ results (Dinelli et al., 2019), but we note thefollowing reasons for not doing a more direct comparison:

1. There are no published Juno/JIRAM H3+ temperatures for Perijove 4 (Feb 2 UTC)
2. Unfortunately, temperatures demonstrably change by tens of Kelvin per hour in the auroral region, so with hundreds K per day of change possible, several days of passage wouldbe difficult to compare to (e.g. Johnson et al., 10.1029/2018JA025511, 2018)
3. While Juno JIRAM records with exceptional spatial resolution, it does so over very narrow swathes of the planet, which makes it difficult to assess regional and global conditions

I would also suggest to discuss the JTGCM model results (Bougher et al. 2005, JGR 110, doi:10.1029/2003JE002230) and compare the observed features to this model. It shows thatthe observed Jovian temperatures and densities in the thermosphere can be satisfactorily matched with a combination of auroral particle and Joule heating at low and high latitudes. We have added a short statement to the final paragraph and now included Bougher et al. (Apr, 2005) as a new summary paragraph reference, note that in the version you received, we had used Majeed et al (2015) as the JTGCM reference. In addition, JTGCM results havenot been replicated since then by several publications (up to 2020 in Yates et al.) so the summary reads "...most thermospheric global circulation models...".

Sentence at line 111 is confusing: temperatures on 25 January are higher inside the main oval with a cold region (T about 650 K) outside the main oval at longitudes 240-330°. Please better describe this part or rephrase.

This has now been clarified as the section has been re-written.

In Figure 3 a and d, the authors show the temperature maps obtained in the two dates. It is quite intriguing the fact that temperature is maximum inside the main oval and in the longitude region 90-240° on 25th January, while on 14th April the maximum temperature value is outside the main oval and at 270-300° longitude. A different behavior seems to occur on the Southern hemisphere during the same date, where there is a hot region at about 150°. It would be interesting to discuss in the text these differences among the two dates.

These last two points are related and highlight that our discussion on this topic was fragmented. We have re-written this section and comment on the morphological differences across the main oval and difference between temperature and density. We have added references to Dinelli et al. (2019) and Johnson et al. (2018) which have studies looking in detail at main auroral oval variability.

Figure 6 (Extended data Figure 3) shows the H₃⁺ temperature as a function of latitude obtained during the two nights of observations. Although it is clear that on 25th January the retrieved temperature is higher than on 14th April for latitudes of 45° equatorward, error bars are missing and so it is not completely evident if the difference is significant. Another possibility would be showing the temperature difference, associated with the error bars superposed to the temperature. This would help in immediately catching the amount of temperature difference.

We agree and have now added error bars which indicate the median uncertainty found at each latitude. In a response to Reviewer 2 we have also followed their suggestion to move this important figure to the main text, as part of Figure 2. Please note that Figure 2 and 3 have also now swapped places, as the plot makes more sense after seeing new Figure 2.

Referee #2 (Remarks to the Author):

General comment

This study addresses the important question of the heat source that maintains the high temperature observed in the Jovian upper atmosphere. This ‘energy crisis’ has been feeding research and controversies in the scientific community for ten of years. It was clearly demonstrated that ultraviolet solar radiation, the main heat source of telluric upper atmospheres is insufficient by hundreds of Kelvin to warm up Jupiter’s thermosphere. One of the most obvious proposed alternatives was that auroral heating through Joule heating and collisional heating in the high-latitude regions was the missing source. This idea was confirmed following the first quantitative observations by HST and the subsequent estimates of the global auroral power deposited in both hemispheres. This rather straightforward deduction was challenged when 3-D Jovian GCMs indicated that the meridional heat transfer to lower latitudes was ineffective owing to the strong Coriolis force and did not provide the heat necessary to explain the observed mid- and low-latitude temperatures.

Thank you for your review. The manuscript title and summary paragraph, which sets the tone for the entire study and details the historical context, has been re-written to make sure we give full context and credit to the earlier work on transport. In the new summary paragraph, amongst the other changes, we have now added important references to Achilleos et al. (1998) and Waite et al. (1983).

The idea that auroral transport could happen is indeed not new, and I can see how our title and summary may have rather brushed over it too quickly. While it was proposed in the past, direct evidence that transport does happen still remains missing. Many alternatives such as gravity waves and acoustic waves continue to be considered a cause of heating, and it is a solution that could work on any planet with an energy crisis. As you have mentioned, most models have been unable to demonstrate auroral transport, so there remains to this day considerable doubt about this very important mechanism. The results of the present study directly demonstrate that transport on a global scale occurs, confirming one of the main hypotheses of earlier work.

In this study, the authors report ground-based spectral measurements of the thermospheric temperature measured on two occasions. They claim that the strong latitudinal temperature gradient they measured and the presence of a region of enhanced temperature separated

from the auroral oval region both demonstrate that the heat transfer from high to low latitudes is efficient.

This is a report of high quality ground-based observations made with the 10-meter Keck II telescope. It clearly shows the presence of an exospheric temperature gradient between the auroral regions and low latitudes in both hemispheres. The importance of auroral heat transfer is further supported by the differences observed between two observation days, one of them possibly concurrent with an increase in the solar wind activity. The data analysis is careful, including a full analysis of the statistics of the measurements. The text is well written and the illustrations are quite clear and well documented.

The statistics is reliable and the error bars are clearly

defined. Thank you.

My main concern is about the novelty and the impact of these results:

In order to alleviate this concern, we have been more concise in the summary paragraph (as discussed above). We also note here that in the three existing Review Chapters available on this topic, each refer to auroral transport as a mechanism of transport that could work, among other mechanisms.

1. Yelle & Miller (2004). Jupiter. The Planet, Satellites & Magnetosphere (Book)
2. Ma et al. (2008). Space Sci. Rev., DOI: 10.1007/s11214-008-9389-1
3. Badman et al. (2015). Space Sci. Rev., doi:10.1007/s11214-014-0042-x

- Unfortunately, no parallel observation of the auroral morphology from HST or Juno could be used to confirm or not an intensification of the auroral precipitation in January 2017 - Previous observations, with coarser spatial resolution, have already indicated that the mid- and low-latitude temperatures are lower than those in the auroral region. See for example, Johnson et al. (JGR, 2018), Moore et al. (GRL, 2017). However, the characteristics of the latitudinal gradient are best seen in Extended data figure 3 of this study

To isolate the dominant heat source acting planetwide a pole to equator map of temperatures at high resolution is essential (based on the points raised earlier).

Johnson et al (2018) provide coverage of the auroral oval, as many studies do, but the measured sub-auroral temperatures have high uncertainties, do not exhibit a gradient and do not extend far beyond the auroral oval.

Moore et al., 2017 show a gradient in some areas close to the auroral oval, but not others, and only extend to 50 degrees north and at coarse spatial resolution. It is important to examine the global case in order to demonstrate the global system, and also to rule out, for example, equatorial heat sources apparently seen in the 1990s (see also our reply to your first specific comment). The present study extends from pole to equator with several thousand data points per map and at a few % uncertainty. We see a snapshot from two different years, possibly representing active and quiet scenarios, allowing the present data to confirm the dominant heat source.

- no mention is made of infrared observations made with the JIRAM on board Juno such as the analysis by Dinelli et al. (2019, Phil. Trans. R. Soc. A 377: 20180406). They offer a spatial resolution between 45 and 130 km.

The present paper is mainly about the global situation, but we agree that the Juno JIRAM observations (with the best resolution ever obtained of the aurora) should be a main part of this body of work, as they offer a valuable insight into the main auroral oval. Ground-

based work benefits from having very good (heavy) instrumentation and global coverage, but it does not have spatial resolution near the poles of Jupiter anywhere less than 1000km/pixel on the planet. We have now added this reference multiple times in our revised discussion of the main auroral oval.

- Many analogies are made with observations or models of Saturn, which has a different behavior in many respects, although they could refer to Jupiter studies instead.

Following this and editorial comments, the second and third paragraphs (which should not include more introduction, according to the editor) have now been removed.

Given these restrictions, I leave it to the editor to decide whether this study should be published in Nature or another journal such as **[redacted]**.

Specific

comments First

paragraph:

l. 9-11 : the lower temperature at low latitude compared to the auroral values is not a new characteristic. It has been observed and discussed before.

We have addressed this in two key areas. Point measurements exist outside of the aurora, but those were not simultaneous, and simultaneity is essential at all latitudes and longitudes because parameters vary greatly between different dates. There has only been one study of global coverage (Lam et al., 1997) using data from 1993. That study has very coarse resolution (with just ~2 pixels between 45-90 North), and shows temperatures near the equator are comparable to the aurora, which would (if accurate) fuel the idea that there are non-auroral heat sources acting at low-latitudes. This has now been added to the main body of the paper in paragraph 2.

To make it clear what is new in the Summary paragraph, we have re-written it.

l. 11-14 : The sentence 'During a period ...region' is very long. I suggest dividing it into two separate sentences.

We agree and have reduced it.

l. 30-32 : Juno has revealed that the source of precipitation may be complex and is generally NOT associated with signatures of field aligned currents (Mauk et al., 2019; Bonfond et al., JGR, 125, e2020JA028152).

Field-aligned currents are no longer mentioned in the paper. As you state, the aurora are much more complicated, so we have taken a more general approach here.

There are many references better suited establishing that electron precipitation heats up the Jovian upper atmosphere.

We now refer to Waite et al. (1983), Achilleos et al. (1998) and Bougher et al. (2005) in the summary paragraph as they refer to heating and the potential for auroral planet-wide heating.

l. 70 : there are many better references in the literature to H₃⁺ density increase in Jupiter's aurora rather than Saturn's case (ground-based, Juno JIRAM).

We agree and have now used the Danelli et al. (2019) and Johnson et al. (2018) references now instead as they are the best known examples

l. 105-108 : How do you determine the 'start time' of the heat wave equatorward propagation?

The start time was determined using the time stamps on the spectra. We have re-worded this section in order to better explain it: “We use the latitude separation between the structure's centre and the main oval, which grows with longitude and therefore with time, since the data are recorded in order of increasing longitude. Equatorward velocities for the hot feature were evaluated between 180 and 260 longitude in steps of 20 longitude, with ~33 minutes of time elapsing between each step due to planetary rotation.”

l. 108-110 : what conclusion do you derive from the discrepancy between the Jovian, Saturnian and terrestrial velocities ?

We simply presented the observation and compared it to other known values to see if it is at least “physical”, but I was surprised how slow Saturnian velocities are, relative to Jupiter and Earth. Referee 3 has suggested that we water down our discussion of this in the paper.

l. 114 : do you mean in the auroral oval ? How do these values compare with column density measurements reported in the literature?

This was a mistake and should have read (which we have now added) that these medians are for the region between 0-30 degrees north latitude. We have also remarked that the 2017 observed column densities are similar to (the only) previous study.

l. 132: this is a very definite statement considering the uncertainties.

This has now been changed to “potentially as a result of” to indicate a level of speculation is involved.

References: some references refer to Saturn's aurora although direct reference to Jupiter's case would be more appropriate.

We have now referenced papers only twice to Saturn (in the last paragraph) which were essential, and we reduced the overall discussion regarding Saturn in the paper. When discussing Jupiter's aurora we reference Dinelli et al. (2019) and Johnson et al. (2019)

Extended data figure 3: this figure appears as a key result of this study and deserves to be included in the main manuscript. If space limitations apply, I would suggest making it an additional panel of Figure 2.

We agree and have added it to Figure 2, and error bars have also been added (per Reviewer 1's suggestion). Please note that Figure 2 and 3 have also now swapped places, as the plot makes more sense after seeing new Figure 2.

Referee #3 (Remarks to the Author):

Nature Review

Title: Global heating of Jupiter's upper atmosphere by auroral energy circulation

Authors: J. O'Donoghue, L. Moore, T. Bhakyaipaul, H. Melin, T. Stallard, J.E.P. Connerney, and C. Tao

This paper presents new observations of H₃⁺ densities and temperatures across the disk of Jupiter. The results clearly indicate temperatures decreasing from the auroral regions toward the equator, strongly suggesting that auroral heating is an important source for Jupiter's

globally high thermospheric temperatures. During the second of two observations, a remarkable detached arc of higher temperatures was observed in the northern hemisphere, centered on an M-shell of ~ 2.5 (approximately the orbit of Amalthea). These data are stunning, and are worthy of publication in Nature on their own. However, the authors includesome speculative interpretations of the data in at least three cases (listed below) which are not justified in the current text. These should either be defended with suitable calculations, or the claims should be watered down a lot.

Thank you for reviewing our paper.

We have watered down the certainty with which we described the hot structure and described the other possible case first (that it is linked to 2.5 M-shell). We removed the notion of Rayleigh drag for Jupiter, as we would rather focus the observational result with the limited space (also, Referee 2 requested less Saturn discussion).

1) One instance of this is in trying to link the arc (and generally higher) temperatures seen in the second observation to solar wind variations, without any real justification as to how the solar wind might influence the upper atmosphere in a region connected to the inner magnetosphere. It has been demonstrated in previous studies that most of Jupiter's auroral activity is internally generated, although solar wind variations certainly play a part in modifying polar auroras. To support their idea that solar wind activity is connected to the high temperatures observed at sub-auroral latitudes, it would be important to demonstrate some physical mechanism linking these two vastly separated regions.

The main justification is that the model demonstration that a solar wind pulse could cause a parcel of heated atmosphere to leave the auroral oval and migrate equatorward (Yates et al., 2014). We have re-written the discussion paragraph in order to more comprehensively explain and justify this feature, but also watered down the language used and began the discussion first with the possibility that the feature is of magnetospheric origin. (more explained below in 2))

2) Another example is the derivation of a meridional speed for the arc feature by assuming that the feature is moving coherently equatorward and began at the main auroral oval. I can understand the interest in deriving something about the evolution of this interesting feature from the snapshot observation, but it seems entirely plausible that it isn't moving at all. The arc feature lies pretty much along a single M-shell, which would be expected if

it had a magnetospheric origin. If it were moving equatorward from the main oval, it seems hard to imagine why the atmosphere it has already passed over at higher latitudes would cool off to the same levels that are seen at lower latitudes that haven't been reached yet.

A magnetospheric origin has now been added as the first possibility for this feature due to its shape closely resembling that of the Amalthean footprint and the discussion was re-written (see also 1) above). We added a key point that the arc is likely to also retain the shape of its origin, so if it came from the main oval it would still have a similar shape to the mapping of Amalthea, explaining its similarity. In long discussions with co-authors before submitting about this topic, it was debated that a significant auroral-like source of plasma would be needed at a remarkably deep region in the inner magnetosphere to explain this feature, and there is no available evidence to show that at present.

Regarding the propagating heat, we have now addressed this with two points. Firstly, our measurements of the main oval may represent a colder, deeper region of atmosphere

because ion production is at lower altitude there, so it may naturally appear colder during highly active precipitation. Secondly, it is also possible that the atmospheric wave was triggered by an individual event that lasted a short time. These points were added to the penultimate paragraph as they relate to the aurora.

3) Finally, the assumption that the Coriolis barrier to equatorward transport can be overcome by Rayleigh friction, as speculated for Saturn in the Müller-Wodarg et al. (2019) paper, is a very interesting idea, but for this paper it should at least be tested for plausibility. The Rayleigh friction acts like viscous drag to damp zonal winds, but winds

in Jupiter's auroral upper atmosphere are observed by Doppler-shifted H₃⁺ emissions to be very large – on the order of km/s. Please provide some estimates for the Rayleigh friction profiles that could allow meridional transport while not shutting down the auroral electrojet. We have removed many references to Saturn now in response to Reviewer 2 and space concerns and tried to focus the paper almost exclusively on Jupiter. As a result, we are saying less about this mechanism at Saturn and how it might work (we have watered down this assumption, in other words). Rather than mention Rayleigh friction, we stick to this being an observational result (it would require a lot of space and modeling to do that topic justice), and simply state that a model for Saturn has presented a possible mechanism that disrupts the trapping of heat in the polar regions. We note to the reviewer, however, that the Rayleigh friction doesn't severely slow Saturn's auroral zonal winds (e.g., from ~1.7 to ~1.5 km/s in the north, and ~1.3 to 1.1 km/s in the south -- Mueller-Wodarg et al., 2019)

If the authors could address these major concerns, then I would support publication in Nature.

Minor comments (mostly to improve readability):

1.2 – “Magnetosphere-atmosphere” -> “On Jupiter, magnetosphere-atmosphere”
The sentence immediately before now says “Jupiter” so I think this is now OK.

1.4 – “global circulation” -> “thermospheric global circulation”
Done

1.6 – “insufficient” -> “weak”
Insufficient is no longer there following the summary paragraph re-write

1.7 – “warm equatorial” -> “warm equatorial
thermosphere”(no longer there)

1.18 – “was recorded by” -> “was observed with”
Done

1.30 – “electrons precipitate” -> “energetic electrons precipitate”

1.31 – “into the planet” -> “toward the planet”

1.32 – “intense auroral emissions” -> “intense auroral emissions through impact excitation of ambient H₂ molecules”

1.33 – “constrast” -> “contrast”

1.34 – “based on solar heating” -> “based on only solar heating”

1.37 – “shown that heat within auroral is” -> “found that heat within the auroral zone is”

1.40 – “altitude” In general, I object that many of the references are to derivative studies, rather than the original work or major reviews. This is just one particular case, but wave heating studies have a long history before this reference.

1.47 – “by the Cassini spacecraft” -> “by Cassini UVIS stellar occultation results”; while discussing other giant planets, you might mention that Neptune also has a large thermosphere temperature, but it has very weak auroras.

The above lines no longer exist, but in response to the comment about referencing derivative studies: we now go further back, referencing Achilleos et al. (1998) and Strobel et al. (1973).

1.66 – “pressure surface” -> “pressure level”
Done

1.73 – “sub-auroral electric current systems” -> “sub-auroral electric current systems (as are common on Earth)”
Done with an added reference to a related Earth paper on the topic (Heelis et al., 2020)

1.107 – “median velocity of 620 ms⁻¹”; as discussed above, if I understand what you’ve done here, it seems that you’ve just measured the latitude difference as a function of longitude between the M-shells of 2.5 and 30. We calculated the peak temperature as a function of latitude at each longitude, which is very close to the M-shells of 2.5 and 30, but we didn’t use the L-shell itself. This section has been re-written to make it more clear how this was found.

1.114 – “The F10.7 index”; does the H3+ auroral brightness vary with F10.7? We assume so yes, but it is superimposed on auroras which are dominated by particle precipitation and heating on short time scales (e.g. even an observing night, Johnson et al., 2018). That is why, with solar EUV ionization being the main production mechanism of H3+ outside of the aurora, we had only commented on the non-auroral parameters. We have made sure to mention earlier that the median column density derivations are for 0-30 degrees north, and have also more efficiently discussed the auroral concerns at the beginning of the paragraph.

Reviewer Reports on the First Revision:

Referees' comments:

Referee #1 (Remarks to the Author):

In the revised version of the paper ‘Evidence of global upper-atmospheric heating at Jupiter by the polar aurorae’ by O’Donoghue et colleagues, the authors improved many points raised in the revision process and answered in a satisfactory manner. I appreciate the mention to other measurements, although non directly comparable to the reported observations, and the mitigation of some very strong statements. I also agree with the changes to the part about the comparison with Saturn. The manuscript is well-written and I think it can be considered for publication.

Referee #2 (Remarks to the Author):

I have read the revised manuscript, compared it to the initial version and to the comments I had

formulated about it.

I find that my questions and suggestions have been adequately addressed. In particular:

- the authors have now included references to earlier work included reporting measurement of thermospheric temperatures.
- they now emphasize the latitudinal variation of the temperature by adding its measured latitudinal distribution in figure 3b.
- the title, summary and text have been modified to reference earlier work suggesting that auroral heating is a likely source of global heating of the Jovian upper atmosphere.
- They also have better justified why these observations have better quality and higher latitudinal resolution than earlier measurements.

Therefore, I can now recommend publication of the manuscript in its present form.

Referee #3 (Remarks to the Author):

I find the authors have addressed my concerns in all respects, and that the paper is suitable for publication in Nature. Congratulations on the amazing data you've collected, and for resisting the temptation to overinterpret them!

Author Rebuttals to First Revision:

N/A